

# Resonating valence bonds and spinon pairing in the Dicke model

**R. Ganesh**[1*]**, L. Theerthagiri**[1]**, G. Baskaran**[1,2]

**1** The Institute of Mathematical Sciences, C I T Campus, HBNI, Chennai 600 113, India
**2** Perimeter Institute for Theoretical Physics, Waterloo, ON, N2L 2Y6 Canada

* ganesh@imsc.res.in

## Abstract

Resonating valence bond (RVB) states are a class of entangled quantum many body wavefunctions with great significance in condensed matter physics. We propose a scheme to synthesize a family of RVB states using a cavity QED setup with two-level atoms coupled to a common photon mode. In the lossy cavity limit, starting with an initial state with $M$ atoms excited and $N$ atoms in the ground state, we show that this setup can be configured as a Stern Gerlach experiment. A measurement of photon emission collapses the wavefunction of atoms onto an RVB state composed of resonating long-ranged singlets. Each emitted photon reduces the number of singlets by unity, replacing it with a pair of lone spins or 'spinons'. As spinons are formed coherently in pairs, they are analogous to Cooper pairs in a superconductor. To simulate pair fluctuations, we propose a protocol in which photons are allowed to escape the cavity undetected. This leads to an inchoate superconductor – mixed quantum state with a fluctuating number of spinon pairs. Remarkably, in the limit of large system sizes, this protocol reveals an underlying quantum phase transition. Upon tuning the initial spin polarization, the emission exhibits a continuous transition from a dark state to a bright state. This opens an exciting route to simulate RVB states and superconductivity.

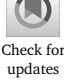
# 1 Introduction

Resonating Valence Bond (RVB) states were originally proposed by Pauling in the context of benzene [1]. They are widely realized in organic chemistry, especially in compounds containing closed loops of carbon atoms. In the arena of condensed matter physics, RVB states have acquired tremendous importance since the discovery of high-$T_c$ superconductivity [2]. The RVB theory of superconductivity [3,4] has given rise to important ideas such as spin liquids [5,6], fractionalization [7], anyonic statistics [8] and topological order [9].

A resonating valence bond state can be defined as a linear superposition of different ways of placing 'dimers' (two-particle singlet states) between pairs of constituent particles (modelled as spins). Benzene provides a simple example – its $\pi$ electrons can form singlets between nearest neighbours in two different ways; the ground state is a symmetric combination of these two (Kekulé) states. RVB theory [3,4] postulates that the undoped cuprates, which are Mott insulators, have an analogous RVB ground state, viz., a superposition of all possible ways to cover the square lattice with singlet dimers. This is an *incompressible* liquid of singlets, e.g., we cannot introduce additional singlets into this state. Unlike benzene, this lattice-RVB contains a very large number of participating states with the number of configurations growing exponentially with system size. Doping removes underlying spins to create 'doublons' and 'holons' [9]. This leads to a compressible singlet fluid which can transport charge – a superconducting state.

Given the richness of RVB states, it would be very useful to realize simple, tunable RVB wavefunctions in the laboratory, with properties that can be evaluated analytically and compared with experiments. Motivated by this goal, we present a cavity-based paradigm to synthesize 'spinon-doped' RVB states. In lattice-RVB systems, an exciting line of investigation has been the response to an applied magnetic field. This is a simpler proposition than conventional doping as it changes the number of singlets without introducing charge dynamics. The field converts singlets ($\frac{1}{\sqrt{2}}\{|10\rangle - |01\rangle\}$) into triplets ($|00\rangle$ or $|11\rangle$). This creates 'spinons' or unpaired spins ($|0\rangle$'s or $|1\rangle$'s) in pairs, imbuing them with a strong tendency to undergo Cooper-like pairing. Condensation of these pairs leads to magnetic order in the plane perpendicular to the field [10–12]. Here, we present a scheme to coherently and controllably introduce spinons

into a parent RVB state. The role of the magnetic field is played by photon emission, via the well known phenomenon of wavefunction collapse. Building on this, we suggest a protocol that simulates spinon pairing arising from Cooper-pair number fluctuations. This gives rise to an incoherent zero-dimensional spinon-superconductor, wherein all spinons are located at the same spatial position.

A quantum phase transition emerges from our analysis of photon emission from the Dicke system. The Dicke model has long been known to host a temperature-tuned (or coupling-tuned) phase transition [13–15]. More recently, several studies have brought out dynamical transitions by including an external drive and dissipation [16–18]. In contrast, our phase transition is non-thermal and non-dynamical in nature as it is driven purely by dissipation. Apart from the intrinsic interest in such a phase transition, it can be exploited to bring the RVB system closer to a coherent superconductor-like pairing state.

This article has three key results. The first is a protocol for a cavity experiment which uses photon detection to synthesize a generalized RVB state. The second is a protocol to simulate spinon pairing by generating fluctuations of unpaired spins. Finally, the third result is a phase transition in the emission properties of the Dicke model which can be used to bring the system closer to superconductivity.

## 2 RVB state from photon detection

### 2.1 Dicke model and photon emission

We consider a cavity QED system with $\mu$ two-level atoms (modelled as $S = 1/2$ spins) coupled to a common photon mode. To allow for a clear detection of photon number, we take the cavity to be in the lossy regime, wherein the rate of photon leakage through a lossy mirror is much higher than the rate associated with spin-photon coupling. The rate of dephasing due to spin-spin interactions is taken to be even smaller and therefore, negligible. We also neglect effects such as non-radiative decay, leakage through the non-lossy mirror, etc. These is precisely the regime in which the recent experiment of Ref. [19] was performed. Our proposal, outlined below, reworks this experiment as a Stern-Gerlach measurement.

Under well-known conditions [19, 20], the spin-photon system is described by the Dicke Hamiltonian within the rotating wave approximation (or more precisely, the Tavis Cummings Hamiltonian),

$$H = B\hat{S}^z_{tot} + \omega a^\dagger a + g\left\{ \hat{S}^-_{tot} a^\dagger + \hat{S}^+_{tot} a \right\}. \tag{1}$$

Here, $B$ is the energy gap between the states of the two-level atom which is assumed to be close to $\omega$, the photon frequency. The spin-photon coupling, $g$, sets the time-scale for photon emission and absorption. This is assumed to be longer than $\kappa$, the rate of photon loss from the cavity. The total spin operator, $\hat{S}^\alpha_{tot} = \sum_{i=1}^{\mu} \hat{S}^\alpha_i$, is the sum of spin operators on all the $\mu$ spins. The Hamiltonian allows for any excited spin to de-excite by emitting into a common photon mode. Likewise, any unexcited spin may absorb a photon and become excited. To a generic state in the Hilbert space of this problem, we can ascribe quantum numbers $S_{tot}$, $m_{tot}$ and $n_{ph}$ – the total spin quantum number, the spin-z quantum number and the number of photons in the cavity, respectively.

Dicke considered the rate of photon emission from an arbitrary initial state by evaluating matrix elements between spin states within a Fermi golden rule approach. He showed that the rate of emission is maximum in a 'superradiant' state with $\{S_{tot} = \mu/2, m_{tot} = 0\}$. In stark contrast, a state with $\{S_{tot} = 0, m_{tot} = 0\}$ is 'dark'. These 'subradiant' states have recently evoked interest as possible quantum memories [21].

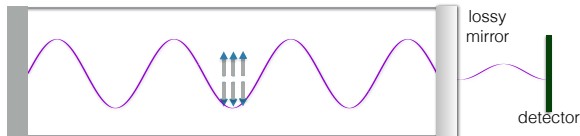

Figure 1: Proposed setup. Spins are initialized in a direct product state and placed inside a cavity with a lossy mirror. The number of photons emitted is measured at the output of the lossy mirror.

It is easy to see that, any state with $\{S_{tot} = \Sigma, m_{tot} = -\Sigma\}$ (assuming $n_{ph} = 0$ is zero) will not radiate as it cannot lower its $m_{tot}$ quantum number any further. This is because the photon creation operator in Eq. 1 is accompanied by the $\hat{S}_{tot}^-$ operator. This is a consequence of the rotating wave approximation which neglects energy non-conserving terms in the spin-photon coupling. More generally, we argue that a state with $\{S_{tot} = \Sigma, m_{tot} = -\Sigma + \nu\}$ will emit precisely $\nu$ photons which can be detected as they leave the cavity. The photon creation operator in Eq. 1 can act on this state precisely $\nu$ times before reaching a non-radiating state. This is an interesting consequence of the lossy cavity limit where the rate of photon loss is higher than the spin-photon coupling. As a result, any emitted photons will leave the cavity before they can be reabsorbed by the spins.

## 2.2 Proposed protocol

In an earlier work, we proposed a protocol to generate dark RVB states by a null-measurement for photon emission [20]. Building on this earlier proposal, we suggest the following protocol to obtain 'spinon-doped' RVBs. The necessary setup is shown schematically in Fig. 1.

- Initialize spins in a direct-product state $|\sigma_1\sigma_2\ldots\sigma_\mu\rangle$, with each $\sigma_i$ being ↑ or ↓, within a cavity with a lossy mirror

- Using a photon detector, count the number of photons emitted via the lossy mirror (the timescale for emission is set by the spin-photon coupling, $g$)

- Measuring the number of photons constitutes a Stern-Gerlach measurement – the spin wavefunction will collapse onto a generalized RVB state

Remarkably, this protocol leads to a generalized RVB state with 'doped' unpaired spins as we discuss below.

## 2.3 Emission from initial state

We assume that spins are initialized in a direct product state as in Refs. [19, 20], with $M$ spins in the excited state and $N$ spins in the ground state. This can be written as

$$|\Psi_{initial}\rangle = |\underbrace{\uparrow\ldots\uparrow}_{M\,\text{spins}}\underbrace{\downarrow\ldots\downarrow}_{N\,\text{spins}}\rangle = \left|\begin{array}{c}\uparrow\ldots\uparrow\\\downarrow\ldots\ldots\downarrow\end{array}\right\rangle. \tag{2}$$

We have arranged the spins in two rows: the 'up' spins in the top row and the 'down' spins in the bottom row. This arrangement brings out the *in-row* permutation symmetry of the initial state. That is, the initial state is invariant under permutations of spins within each row. We note that this state has $m_{tot} = (M-N)/2$. From elementary principles of angular momentum

addition, it can be written as a superposition of states with $S_{tot} = |N-M|/2, \ldots, (N+M)/2$,

$$|\Psi_{initial}\rangle = \sum_{S=\frac{|N-M|}{2}}^{\frac{N+M}{2}} a_S |S_{tot} = S, m_{tot} = \frac{M-N}{2}\rangle$$

$$= \sum_{\nu=\nu_{min}}^{M} a_\nu |S_{tot} = \frac{N-M}{2} + \nu, m_{tot} = \frac{M-N}{2}\rangle. \tag{3}$$

The lowest value of the summation index, $\nu_{min}$, is 0 if $M < N$ and $(M-N)$ if $M > N$. We see that each component in this linear superposition is of the form $\{S_{tot} = \Sigma, m_{tot} = -\Sigma + \nu\}$. From our earlier arguments, such a state will emit precisely $\nu$ photons. Thus, each component in Eq. 3 will emit a different number of photons.

As in the Stern Gerlach experiment, a measurement of emitted photon number will collapse the spin wavefunction onto the corresponding $S_{tot}$ sector. The possible outcomes for emitted photon number are $\nu_{min}, \ldots, M$. If $p$ photons are detected, the spin state collapses onto

$$|\Psi_p^{M,N}\rangle \sim \left\{\hat{S}_{tot}^-\right\}^p \hat{P}_{S_{tot} = \frac{N-M}{2} + p} |\Psi_{initial}\rangle. \tag{4}$$

This can be summarized as follows: The projection onto the subspace with $\left\{S_{tot} = \frac{N-M}{2} + p\right\}$ picks out the correct component from Eq. 3. To emit $p$ photons, the spin system must lower its $S_z$ quantum number by $p$; this is accompanied by the $\left\{\hat{S}_{tot}^-\right\}^p$ operator. The resulting state, duly normalized, is the wavefunction obtained from the above protocol when $p$ photons are detected.

### 2.4 RVB state from wavefunction collapse

The state obtained from wavefunction collapse, $|\Psi_p^{M,N}\rangle$ of Eq. 4, is an RVB state that can be written down using the following simple set of rules. In Appendix A, we give an explicit proof that the collapsed wavefunction indeed takes this form.

The construction of the RVB state is shown in Fig. 2. We first place $(M-p)$ dimers, each connecting a spin from the top row to one in the bottom row. The constituent spins may be arbitrarily chosen with the condition that no spin can be part of more than one dimer. Each dimer here denotes a singlet wavefunction $\{|\uparrow_t\downarrow_b\rangle - |\downarrow_t\uparrow_b\rangle\}/\sqrt{2}$. The singlet wavefunction is always 'ordered', i.e., the spin from the top row appears first. This state has $p$ unpaired spins in the top row and $(N-M+p)$ unpaired spins in the bottom row. Note that $(N-M+p)$ is non-negative even if $M > N$, as $p$ is then constrained to $\{(M-N), \ldots, M\}$. All the unpaired spins are assumed to point down. We now perform all possible permutations of the top row as well as the bottom row, and linearly superpose all such states. After normalization, this results in an RVB state which is immediately seen to have in-row permutation symmetry.

## 3 Simulating spinon superconductivity

### 3.1 Emission induced doping of RVB state

The state obtained due to detection of $p$ photons is a generalized RVB state with $(N-M)+2p$ unpaired spins. The possible outcomes for photon number measurement are $p = 0, \ldots, M$ if $M < N$ and $p = (M-N), \ldots, M$ if $M > N$. Each outcome collapses the wavefunction onto a particular generalized RVB state. This is depicted in Fig. 3 for two cases, $(M=2, N=2)$ and $(M=2, N=3)$. The collapsed states differ in the number of unpaired spins. However, for

$$|\Psi_{RVB}\rangle = \frac{1}{\delta_{M,N,p}} \sum_{P_M} \sum_{P_N} \left| \overbrace{\bigcirc \cdots \bigcirc}^{\substack{M-p \\ \text{dimers}}} \overbrace{\downarrow \cdots \downarrow}^{\substack{p \\ \text{spins}}} \underbrace{\downarrow \cdots \downarrow}_{\substack{N-M+p \\ \text{unpaired spins}}} \right\rangle$$

Figure 2: RVB state after emission of $p$ photons. The sum over $P_M$ denotes a sum over all permutations of the $M$ spins in the top row, with $M!$ permutations. The sum over $P_N$ denotes a sum over all $N!$ permutations of the $N$ spins in the bottom row. There is an overall normalisation constant $\delta_{M,N,p}$.

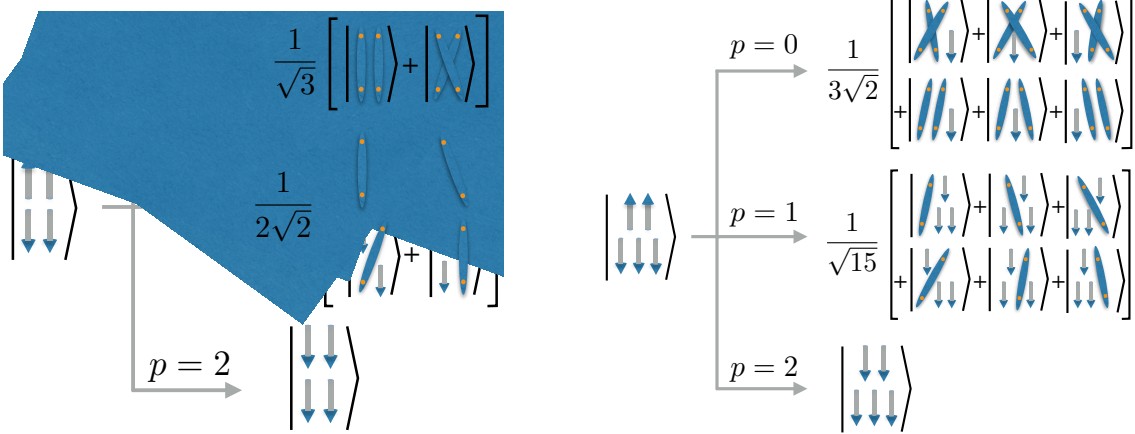

Figure 3: RVB states arising from photon measurement. Left: For the initial state with $M = 2$ and $N = 2$, possible outcomes for number of photons emitted are $p = 0, 1, 2$. Each outcome collapses the spin wavefunction into an RVB state which contains 0, 2 and 4 unpaired spins respectively. Right: For the initial state with $M = 2$ and $N = 3$, possible outcomes for photon emission are $p = 0, 1, 2$. The collapsed RVB states contain 1, 3 and 5 unpaired spins respectively.

a given initial state, the number of unpaired spins is either always even or always odd. This is a simple example of 'topological order'. This can be rephrased as follows: The detection of photons collapses the spin wavefunction, creating one pair of unpaired spins for every photon observed.

To illustrate this, let us consider the case of $M = N$. In this case, the possible outcomes for photon number are $p = 0, \cdots, M$. When no photon is observed, the collapsed wavefunction is an RVB state with $S_{tot} = 0$ containing $M$ dimers and no unpaired spins. This 'strong' RVB state was discussed in our earlier work [20] where a similar protocol was proposed to isolate subradiant (non-emitting) states. Here, we have extended this idea to non-zero emission, showing that a positive detection of photons also leads to an RVB state. For example, if one photon is detected, we obtain a modified RVB state with two unpaired spins. This state emerges by breaking one dimer of the strong RVB state. Similarly, detection of two photons leaves us with four unpaired spins, obtained by breaking two dimers. Proceeding in this way, each emitted photon breaks a dimer. When we see maximal emission of $M$ photons, we have $2M$ unpaired spins with every dimer of the strong RVB broken. This is illustrated in Table. 1.

Table 1: Collapse by photon detection for $M = N$. The columns show $p$ (the number of photons detected), the $S_{tot}$ sector onto which collapse occurs, and the number of unpaired spins in the resulting RVB state.

| $p$ | $S_{tot}$ | Unpaired spins |
|---|---|---|
| 0 | 0 | 0 (strong RVB) |
| 1 | 1 | 2 |
| 2 | 2 | 4 |
| $\vdots$ | $\vdots$ | $\vdots$ |
| N | N | 2N (all $\downarrow$) |

Photon measurement plays a role here that is analogous to a magnetic field in lattice-RVB systems [10]. It creates unpaired spins, which are explicitly seen to be created in pairs. In lattice systems, these 'spinon' excitations are argued to have fermionic statistics [9]. In contrast, we consider a zero-dimensional system. As seen from the Hamiltonian in Eq. 1, all spins are effectively at the same position with regard to the spin-photon coupling. As there is no room for particle exchange, exchange statistics is irrelevant. Nevertheless, we show below that a superconductor-like pairing can be induced between spinons.

### 3.2 Non-measurement of photons and superconductivity

The connection between RVB states and superconductivity can be brought out by a modified protocol with a setup as shown in Fig. 4:

- Synthesize a direct-product state $|\underbrace{\uparrow \ldots \uparrow}_{M\,\mathrm{spins}} \underbrace{\downarrow \ldots \downarrow}_{N\,\mathrm{spins}}\rangle$ inside a cavity with a lossy mirror

- Allow photons, if emitted, to escape with no detector at the output of the lossy mirror

In the lossy cavity limit, this protocol corresponds to tracing over the photon degrees of freedom. After a long enough waiting time (time scale set by spin-photon coupling), any photons generated by the spins would have left the cavity. The spin system is now described by a density matrix given by

$$\hat{\rho}_{sp} = \mathrm{Tr}_{ph}\hat{\rho}_{sp-ph}, \tag{5}$$

where $\hat{\rho}_{sp-ph}$ is given by

$$\left[ \sum_{p=p_{min}}^{M} |\Psi_p^{M,N}\rangle \otimes |p\rangle_{ph} \right] \left[ \sum_{p'=p_{min}}^{M} \langle\Psi_{p'}^{M,N}| \otimes \langle p'|_{ph} \right].$$

The components of the combined spin-photon wavefunction are indexed by $p$, the emitted photon number. The spin wavefunction $|\Psi_p^{M,N}\rangle$ is the RVB wavefunction shown in Fig. 2 with $(N - M) + 2p$ unpaired spins. After tracing over photons, we obtain a reduced density matrix for spins with each component having a different number of unpaired spins. This is a *mixed* state as there is no fixed phase relationship between sectors with different numbers of unpaired spins.

Remarkably, this is an analogue of a finite-sized superconductor. It contains a fluctuating number of unpaired spins within either the odd or the even sector; in other words, it contains

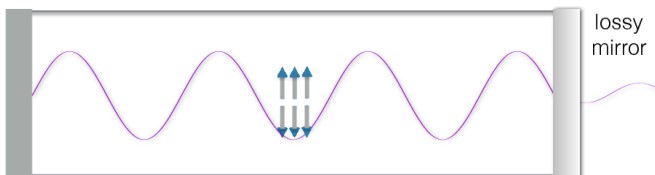

Figure 4: Modified setup without measuring photon output. This is equivalent to tracing over photon degrees of freedom.

a fluctuating number of spinon pairs. This is to be compared with a superconductor which is essentially a Bose condensate of Cooper pairs, i.e., a linear superposition of components with varying pair-numbers. This superposition is 'coherent', i.e., the components have amplitudes with well-defined phase differences. This is a signature of spontaneous breaking of $U(1)$ gauge symmetry which occurs in the thermodynamic limit. In a finite-sized system, however, spontaneous symmetry breaking is forbidden. While this preempts condensation into a coherent state, we will nevertheless have fluctuations in Cooper pair number which remain incoherent. This is precisely the character of the density matrix obtained here – i.e., our protocol creates an analogue of an incoherent superconductor. In the next section, we characterize the distribution of pair numbers to examine the proximity to a coherent distribution. In the process, we uncover a phase transition in the Dicke model.

## 4 Phase transition in photon emission

### 4.1 Relative probabilities for photon emission

Dicke, in his 1954 paper, argues that photon emission is dominated by a 'superradiant' state with maximal $S_{tot}$ and $m_{tot} = 0$. On this basis, radiation properties are often studied by neglecting non-superradiant states, retaining only states with maximal $S_{tot}$ [22,23]. However, it is important to note that Dicke's assertion is a statement about the rate of emission (or absorption) of a single photon. In other words, it determines the state which is the fastest to emit one photon, irrespective of whether or not more photons will follow. This definition of superradiance is not relevant in situations such as the protocol described above. Here, it is more appropriate to ask the following question. Given an initial state $|\Psi_{initial}\rangle$ as defined in Eq. 3, what is the probability distribution of the number of emitted photons?

The limiting cases of this probability distribution can be easily deduced. For $M < N$, we can have null emission ($p = 0$). The corresponding probability was calculated in Ref. [20], with $P_{p=0} = (N - M + 1)/(N + 1)$. At the other extreme, the probability for emission of $M$ photons (maximum emission) can be calculated as follows. Its probability amplitude is given by

$$a_M = \langle \Psi_{initial} | \Psi_{S_{tot}=\frac{N+M}{2}, m_{tot}=\frac{M-N}{2}} \rangle. \tag{6}$$

Here, $|\Psi_{S_{tot}=\frac{N+M}{2}, m_{tot}=\frac{M-N}{2}}\rangle$ is a 'superradiant' state as it has maximal $S_{tot}$ for the system of $(N + M)$ spins. As superradiant states are fully symmetric under all permutations, this state can be explicitly written down as

$$|\Psi_{S_{tot}=\frac{N+M}{2}, m_{tot}=\frac{M-N}{2}}\rangle = \binom{N+M}{M}^{-1/2} \sum_{C_{M,N}} |\underbrace{\uparrow \ldots \uparrow}_{M\,\text{spins}} \underbrace{\downarrow \ldots \ldots \downarrow}_{N\,\text{spins}}\rangle. \tag{7}$$

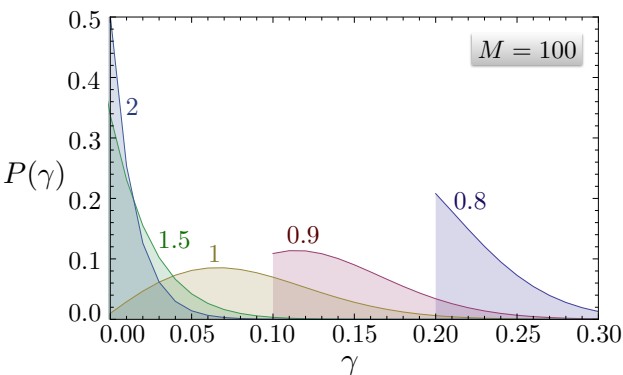

Figure 5: Probability distribution of number of photons emitted, $P(\gamma)$ vs. $\gamma$. This distribution is shown for different $\alpha$, which denotes the imbalance in the initial state. The $\alpha$ corresponding to each curve is shown.

Here, the summation is over $\begin{pmatrix} N + M \\ M \end{pmatrix}$ possible rearrangements of spins. From this explicit form of the superradiant wavefunction, we obtain $a_M = \begin{pmatrix} N + M \\ M \end{pmatrix}^{-1/2}$. The probability for emission of $M$ photons is simply the square of this quantity. This acquires a simple form when $M = N$ and $M \gg 1$. Using Stirling's approximation, we obtain $P_{N=M}(M) \sim 2^{-2M}$. We see that, when $N = M$, the probability for superradiant (maximal) emission is exponentially small! This is in stark contrast to a naive reading of Dicke's result which would suggest that superradiant states dominate emission. Below, we look at the profile of emission from different initial states and show that it is generically dominated by states far from the superradiant limit. This is reflected in the superconductor-like nature of the spin state left behind after emission.

## 4.2 Tuning the imbalance in the initial state

We take an initial state with $M$ up-spins and $N$ down-spins. Upon tuning the imbalance between up- and down-spins, we find a 'phase transition' in the emission properties. To see this, we treat $M$ as the tuning parameter controlling system size with $M \to \infty$ being the thermodynamic limit. We define $\alpha = N/M$ and $\gamma = p/M$, where $p$ is the number of photons emitted. The parameter $\alpha$ quantifies the imbalance in the initial state: $\alpha = 0$ represents a superradiant initial state with all spins pointing up, while $\alpha = 1$ is the balanced initial state with equal numbers of up- and down-spins. As for photon emission, the maximum number that can be emitted is $M$ as we initially have $M$ up-spins. The parameter $\gamma$ represents photon emission as a fraction of this maximum.

In Fig. 5, we plot the probability distribution for photon emission, $P(\gamma)$ vs. $\gamma$, for various $\alpha$ values with $M = 100$ fixed. Note that the area under the $P(\gamma)$ curve must be $1/M$ (i.e., $\int_0^1 d\gamma P(\gamma) = 1/M$) on account of the spacing in allowed $\gamma$ values. There are two limiting cases:

- $\alpha = 0$: all spins are initially up. In this case, $M$ photons will escape from the cavity. The probability distribution $P(\gamma)$ becomes a delta function (with weight $1/M$) centred at $\gamma = 1$.

- $\alpha \to \infty$: all spins are initially down. In this case, no photon will be emitted with $P(\gamma)$ being a delta function centred at $\gamma = 0$.

As we tune $\alpha$ between these limits, keeping $M$ fixed at a finite value, we see that $P(\gamma)$ smoothly evolves acquiring a bell-like shape at a range of intermediate $\alpha$ values. Note that, for $\alpha < 1$, at least $(M-N)$ photons will be emitted (see discussion following Eq. 3); as a consequence, the $P(\gamma)$ curve begins abruptly at $\gamma_c = 1 - \alpha$.

### 4.3 Photon distribution in the thermodynamic limit

In the thermodynamic limit, the photon probability distribution $P(\gamma)$ takes a simple form. Given the initial state of Eq. 3, the probability of emission of $p$ photons can be expressed in terms of Clebsch Gordan coefficients,

$$P_{N,M}(p) = |a_p|^2 = |C(j_1, m_1, j_2, m_2, j_3)|^2, \tag{8}$$

where $a_\nu$ are the coefficients in Eq. 3. This is identified as a Clebsch Gordan coefficient with $j_1 = M/2$, $m_1 = M/2$, $j_2 = N/2$, $m_2 = -N/2$ and $j_3 = (\frac{N-M}{2} + p)$. This identification stems from recasting Eq. 3 as angular momentum addition. In the initial state, the top row forms a net moment with $S = M/2$ and $m = M/2$ while the bottom row forms a net moment with $S = N/2$ and $m = -N/2$. The sum of these two moments is resolved into $S_{tot}$ components in Eq. 3. The Clebsch Gordan coefficients take a particularly simple form [24, 25] to give

$$P_{N,M}(p) = \frac{M!N!(1 + 2p + N - M)}{(M-p)!(1 + N + p)!}. \tag{9}$$

We now express this in terms of our parameters $\alpha$ and $\gamma$. After a few simple manipulations (see Appendix B), assuming $\gamma, \alpha \gg 1/M$ (focussing on the regime where neither imbalance nor emission is negligible), we obtain

$$P_\alpha(\gamma) \approx \frac{2\gamma + \alpha - 1}{\alpha + \gamma} \exp\left[ M \int_0^\gamma d\zeta \left( \ln\{1 - \zeta\} - \ln\{\alpha + \zeta\} \right) \right]. \tag{10}$$

This can be reduced to a convenient form when $\alpha$ is close to unity, i.e., for small values of imbalance (see Appendix B for detailed derivations). In this regime, we find that the emission is very low, i.e., $P(\gamma)$ is non-negligible only when $\gamma$ is close to zero. We consider three separate cases which are shown schematically in Fig. 6:

- $\alpha > 1$: This represents an imbalance with greater number of down-spins in the initial state. In this case, we find that the probability distribution $P(\gamma)$ is peaked at $\gamma = 0$. It can be approximated as

$$P_{\alpha>1}(\gamma) \approx (\alpha - 1)\exp[-M(\alpha - 1)\gamma]. \tag{11}$$

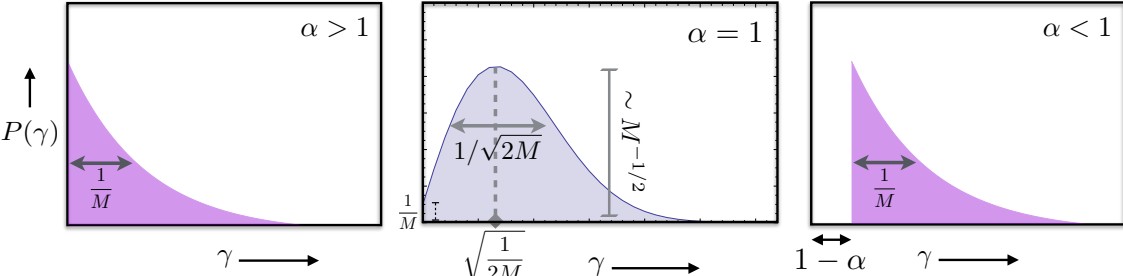

Figure 6: Probability distributions for photon emission in the thermodynamic limit. Left: Imbalanced case with more down-spins in the initial state ($N > M$). Centre: Balanced case with equal numbers of up- and down-spins initially ($M = N$). Right: Imbalanced case with more up-spins initially ($M > N$).

Thus, the distribution decays exponentially with a width that scales as $M^{-1}$. As $M \to \infty$, this distribution approaches a delta-function centred at $\gamma = 0$. In other words, the net emission goes to zero and the state becomes 'dark'.

- $\alpha < 1$: This is an imbalance with greater number of up-spins in the initial state. In this case, atleast $(M - N)$ photons will be emitted; as a consequence, the distribution is uniformly zero when $\gamma < \gamma_c$, where $\gamma_c = 1 - \alpha$. For $\gamma > \gamma_c$, the probability distribution can be approximated as

$$P_{\alpha<1}(\gamma) \approx P_0 \exp\left[-3M(1-\alpha)(\gamma - \gamma_c)\right]. \tag{12}$$

Beyond $\gamma_c$, the distribution decays exponentially with the width scaling as $M^{-1}$. As $M \to \infty$, this distribution approaches a delta-function at $\gamma = \gamma_c$. In the thermodynamic limit, precisely $(M - N)$ photons are emitted, which can be much smaller than $M$, the maximum possible photon number.

- $\alpha = 1$: This is the balanced case with equal numbers of up- and down-spins in the initial state. The probability distribution can be approximated as

$$P_{\alpha=1}(\gamma) \approx \frac{2\gamma}{1+\gamma} \exp\left[-M\gamma^2\right]. \tag{13}$$

As shown in Fig. 6(centre), this distribution is non-monotonic with a maximum at $\sim 1/\sqrt{2M}$. Around this peak, it has a bell shape with width proportional to $1/\sqrt{M}$. As we approach $M \to \infty$, the location of the peak as well as the width of the distribution go to zero. This is a remarkable property: the balanced state, in the thermodynamic limit, becomes 'dark' and traps all excitations.

In all the three cases above, as long as $\alpha$ is not too small (we are not too close to the superradiant initial state with all spins up), the emission is dominated by states far from the superradiant ($\gamma = 1$) limit.

To see the dependence of emission on imbalance, we plot the expectation value of photon emission, $\bar{\gamma}$, defined as $\bar{\gamma}_\alpha = \sum_{\gamma=0}^{1} \gamma P_\alpha(\gamma)$. Here, the sum is over all possible values of photon emission parametrized by $\gamma$. As $\gamma$ is defined as $p/M$, it increases in steps of $1/M$. We use the exact probability distribution $P_\alpha(\gamma)$, derived from Eq. 9. The quantity $\bar{\gamma}_\alpha$ is to be understood as $\bar{p}_{M,N}/M$, where $\bar{p}_{M,N}$ is the expectation value of the number of photons emitted from an initial state with $M$ up-spins and $N$ down-spins. The obtained values of $\bar{\gamma}_\alpha$ are plotted as a function of imbalance $\alpha$ in Fig. 7 for various system sizes, $M$. The extrapolated value for $M \to \infty$ is shown as a dashed line. Remarkably, this extrapolated value vanishes for all $\alpha \geq 1$. For $\alpha < 1$, this value increases linearly and approaches unity at $\alpha = 0$. As $\alpha = 0$ is the superradiant limit, we indeed expect maximal photon emission, i.e., $\bar{p}_{M,0} = M$, or equivalently, $\bar{\gamma}_{\alpha=0} = 1$.

This plot clearly reveals a continuous phase transition. The tuning parameter for this transition is $\alpha$, the imbalance in the initial state. The order parameter which reveals the transition is $\bar{\gamma}_\alpha$, the expectation value of emission. On the disordered side of the transition ($\alpha \geq 1$), the order parameter vanishes signalling 'darkness'. On the ordered side, it is non-zero and steadily increases. This indicates that a net emission of photons develops and steadily increases to reach the maximal value in the superradiant limit of $\alpha = 0$. The critical point that separates the two phases is $\alpha = 1$, the balanced case. Remarkably, this critical balanced initial state does not radiate in the thermodynamic limit!

The presence of a phase transition is also reflected in the variance of $\gamma$ (or equivalently, in the variance of $p$, the number of emitted photons). In Fig. 7 (bottom), we plot $\{M \times (\Delta\gamma)_\alpha\}$ vs. $\alpha$, where $(\Delta\gamma)_\alpha$ is the variance of $\gamma$ calculated using the probability distribution $P_\alpha(\gamma)$ obtained from Eq. 9. As we increase $M$, we see that the variance becomes sharp. This can be

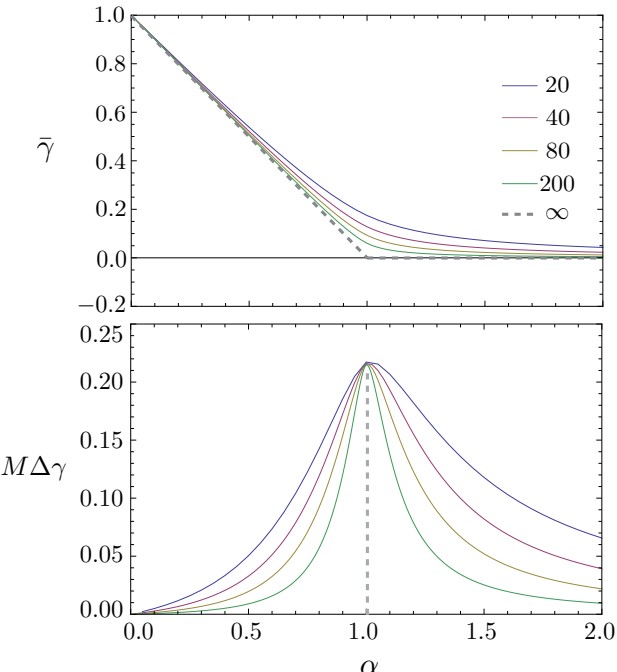

Figure 7: Top: Average photon emission ($\bar{\gamma}$) vs. imbalance $\alpha$, for various system sizes ($M$). The inferred curve in the thermodynamic limit ($M \rightarrow \infty$) is shown as the grey dashed line. Bottom: Variance of $\gamma$ multiplied by system size $M$, as a function of $\alpha$. The plot is non-monotonic with a peak at $\alpha = 1$. In the thermodynamic limit, this plot becomes a delta function.

understood from the spread of the distribution as shown for three cases in Fig. 6. When $\alpha \neq 1$, the standard deviation scales as $1/M$ and the variance scales as $1/M^2$. As a result, $\{M \times (\Delta\gamma)_\alpha\}$ vanishes as $M \rightarrow \infty$. In contrast, at the critical point given by $\alpha = 1$, the variance scales as $1/M$ and $\{M \times (\Delta\gamma)_\alpha\}$ approaches a constant value.

We have demonstrated that the emission of photons from a direct-product initial state shows a phase transition. This is a *quantum* phase transition as there is no temperature scale involved in the problem. Indeed, it is initial imbalance, $\alpha$, that is the tuning parameter. The transition is seen in the steady state density matrix after all possible emission has taken place – in this sense, this is a transition driven by dissipation alone. It indicates a new route to phase transitions in open quantum systems.

### 4.4 Consequences for superconductivity

The phase transition in photon emission has important consequences for the simulation of spinon superconductivity. In earlier sections, we demonstrated that the measurement of emission collapses the spin wavefunction into an RVB state with $(N - M + 2p) = M(\alpha - 1 + 2\gamma)$ unpaired spins. Building on this, we showed that emission without photon measurement leads to a mixed spin state which has fluctuating number of unpaired spins. Denoting the number of unpaired spins by $q$, its expectation value is given by $\bar{q} = N - M + 2\bar{p} = M\{\alpha - 1 + 2\bar{\gamma}_\alpha\}$. In the same way, the variance of the number of unpaired spins is $\Delta q = 4M^2(\Delta\gamma)_\alpha$ with no explicit dependence on $\alpha$. We have shown how $\bar{\gamma}_\alpha$ and $(\Delta\gamma)_\alpha$ vary with $M$ and $\alpha$ above. These two parameters can also be used to control the distribution of unpaired spins left behind after photon loss. By carefully choosing the two parameters, the distribution of unpaired spins can be tuned from sub-Poissonian to super-Poissonian regimes. For example, in the balanced $\alpha = 1$ case, we have $\bar{q} \sim M^{1/2}$ while $\Delta q \sim M$. For large $M$, the mean is smaller than the variance

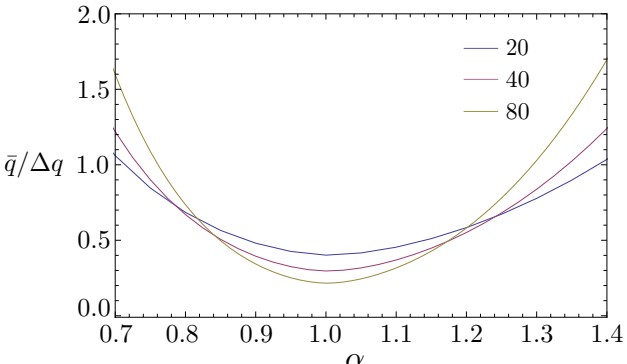

Figure 8: Characterizing the distribution of unpaired spins, $q$, left behind after emission. The ratio of the mean to the variance as a function of imbalance ($\alpha$), for three different system sizes ($M$).

leading to super-poissonian character. With a non-zero imbalance, beyond a threshold system size, the distribution becomes sub-poissonian with the mean being smaller than the variance. Fig. 8 shows the ratio of the mean number of unpaired spins to its variance for different system sizes. Thus, by tuning $\alpha$ and $M$, the distribution of unpaired spins can be altered. By tuning across a Poissonian distribution, we can increase the tendency towards coherent Cooper pair formation.

## 5   Summary and Discussion

We have presented a scheme to synthesize RVB states and to simulate spinon-doping in a cavity-QED experiment. We have presented three key results: (i) a Stern-Gerlach measurement of emitted photons will collapse spins into a spinon-doped RVB state. This proposal is in line with the recent thrust to use cavity based systems to prepare entangled many body wavefunctions [26, 27]. (ii) The non-measurement of photons leads to a mixed state with a fluctuating number of spinon pairs, an analogue of a finite sized superconductor. (iii) Our protocol with non-measurement of photons, when extended to large system sizes, reveals a new phase transition in the open Dicke model. By tuning imbalance in the initial state, we see a continuous transition from dark to bright character. This suggests a new route to study phase transitions and criticality in open systems.

Our proposal using a Dicke system has several advantages over solid state experiments, as it allows for the precise synthesis and characterization of RVB states. It allows for greater tunability than similar proposals in cold atom systems [28, 29]. For example, our approach allows for systematically increasing system size to study the approach to spontaneous symmetry breaking (condensation of Cooper pairs), an interesting problem in its own right [30]. At every step, experimental results can be compared with analytic calculations using the explicit wavefunctions that we have provided.

There are a few key assumptions in our analysis:

(a) The rotating wave approximation allows us to neglect energy-non-conserving terms of the form $\hat{S}_{tot}^{+} a^{\dagger}$ and $\hat{S}_{tot}^{-} a$ in Eq. 1. This is expected to be valid when the spin-photon coupling is much smaller than the frequency of the cavity mode.

(b) Spin dissipation (non-radiative decay and dephasing) may be neglected. With such terms, $S_{tot}$ would no longer be a conserved quantity.

(c) The lossy cavity limit allows for emitted photons to leave the cavity for detection and to not

be reabsorbed. This requires the rate of photon loss to be much greater than the spin-photon coupling.

(d) The waiting time for photon emission remains reasonably short as the system size increases. This is a crucial requirement to make precise measurements of emitted photon number.

(e) Spatial variations in the spin-photon coupling are negligible. This may not hold for large system sizes with many atoms within the cavity.

(f) Spin-spin interactions may be neglected.

With a suitable experimental system such as that in Ref. [19], these assumptions can be justified. This experiment was performed using superconducting qubits in a microwave cavity [19]. In this experiment, the resonance frequency ($\sim 7.06$ GHz) was much larger than the spin-photon coupling (about 3.5 MHz), in accordance with (a) above. In turn, the spin-photon coupling was much greater than the non-radiative decay (0.04 MHz) and the dephasing rate (0.25 MHz), in line with (b). In contrast, the spin-photon coupling was much weaker than the cavity loss rate (43 MHz), as required by (c). Similar parameters are potentially achievable in other systems such as cold gases [15,31], ion traps [32], diamond with nitrogen-vacancies [33] and nuclear ensembles [34]. RVB character of collapsed states can be ascertained by measuring spin-spin correlations, which can be worked out from our explicit wavefunctions.

Small violations of our assumptions, as will occur in any experimental setup, can be taken into account using a master-equation based approach [35, 36]. This approach can be used to test if our predictions such as RVB character, emission phase transition, etc. survive. It can also give quantitative estimates for the photon emission time and place bounds on the assumption (d). This is beyond the scope of this work and will be taken up in the future.

A weak spin-spin interaction may serve as a useful tool here to induce coherence across sectors with different Cooper pair numbers. If such an effect exists, it will manifest as coherence in the emitted photon output. In particular, it will lead to coherence between different photon number sectors. The recent experiment of Mlynek et. al. [19] measured the density matrix of emitted photons. Similar measurements, when extended to bigger systems, may show coherence between photon number sectors. This may point to connections between lasing and RVB theory, suggesting an interesting future direction. Relaxing the assumptions (e) and (f) will introduce variations of the coupling constant in space and take us beyond the Dicke model. Suitable engineering of the coupling can be used generate short-ranged RVB states. This can be used to simulate spin liquid states with topological order [37].

We have demonstrated a quantum phase transition in the emission from an open Dicke system. It is driven by singular behaviour of the multiplicity function (the distribution of weight among different $S_{tot}$ values), reminiscent of the Ising ferromagnet [38]. The transition is deeply connected to the RVB character of wavefunctions. For example, we have shown that the balanced initial state with equal numbers of up- and down-spins is dark in the thermodynamic limit. This is because of the dominant weight coming from the $S_{tot} = 0$ sector, which retains a finite weight even in the thermodynamic limit. This component is, as we have shown above, an RVB state with the maximum number of singlet dimers, a superposition of a very large number of 'dimer covers'. It is the large number of these component states that increases the weight of this sector. Indeed, in all imbalanced initial states, we see that the dominant component in the thermodynamic limit is the RVB state containing the largest number of singlet dimers. This shows the strong tendency towards singlet formation, the role of resonance between valence bond configurations, and the utility of the RVB description.

## Acknowledgements

We thank Manas Kulkarni and B. Prasanna Venkatesh for useful discussions. LT thanks Science and Engineering Research Board (SERB, India) for financial support. GB thanks SERB, India for a SERB Distinguished Fellowship.

**Funding information**   Research at Perimeter Institute is supported by the Government of Canada through the Department of Innovation, Science and Economic Development Canada and by the Province of Ontario through the Ministry of Research, Innovation and Science.

## A   RVB nature of collapsed state

We provide an explicit proof for the RVB nature of the collapsed state here. We first reexpress the collapsed state using a row wise decomposition. We then show that the postulated RVB state has the same decomposition, thereby showing that the states are identical.

### A.1   Row wise decomposition of collapsed state

In the main text, we have an expression for the wavefunction obtained by collapse due to observation of $p$ photons. This collapsed state $|\Psi_p^{M,N}\rangle$ has a high degree of symmetry – it is invariant under in-row permutations, a symmetry inherited from $|\Psi_{initial}\rangle$. This can be seen from Eq. 4 wherein the projection operator $\hat{P}_{S_{tot}=\frac{N-M}{2}+p}$ and the spin lowering operator $\left\{\hat{S}_{tot}^-\right\}^p$ are symmetric under all permutations of the $(N+M)$ constituent spins [20]. We use this property to reexpress $|\Psi_p^{M,N}\rangle$ in a convenient form below.

We expand $|\Psi_p^{M,N}\rangle$ in terms of row wavefunctions,

$$
\begin{aligned}
|\Psi_p^{M,N}\rangle = \sum_{\lambda=p}^{M} E_\lambda |S_{tot}=M/2, m_{tot}=M/2-\lambda\rangle_t \otimes \\
|S_{tot}=N/2, m_{tot}=\lambda-N/2-p\rangle_b.
\end{aligned}
\tag{14}
$$

Here, the subscripts $t$ and $b$ denote top and bottom row wave functions. To obtain this form, we have used the powerful constraint of in-row permutation symmetry. This forces each row wavefunction to have the maximal possible $S_{tot}$ value [20]. As a consequence, the top and bottom rows have $S_{tot}=M/2$ and $N/2$ respectively. The $m_{tot}$ quantum numbers on the right hand side of Eq. 14 are chosen to reproduce the $S_z$ quantum number of $|\Psi_p^{M,N}\rangle$ which is $(\frac{M-N}{2})-p$.

The above Eq. 14 provides a Schmidt decomposition of $|\Psi_p^{M,N}\rangle$ in terms of row-wavefunctions. We now see that the left hand side of Eq. 14 is a wavefunction of $(N+M)$ spins with $\left\{S_{tot}=\frac{N-M}{2}+p, m_{tot}=\frac{M-N}{2}-p\right\}$. The right hand side is a combination of spins with $S_{tot}=M/2$ and $N/2$. This equation signifies usual angular momentum addition where $E_\lambda$ are Clebsch Gordan coefficients. In particular, we identify $E_\lambda = C(j_1 j_2 J; m_1 m_2)$ with $j_1=M/2$, $m_1=M/2-\lambda$, $j_2=N/2$, $m_2=\lambda-N/2-p$ and $J=\frac{N-M}{2}+p$. Using known expressions for Clebsch Gordan coefficients [24], we obtain $E_\lambda$ as

$$
\begin{aligned}
E_\lambda = {} & \frac{(-1)^{\lambda-p}(N+p-\lambda)!\lambda!}{(N-M+p)!p!}(N-M+2p+1)^{1/2} \times \\
& \left[\frac{p!(N-M+p)!(M-p)!(N-M+2p)!}{(N+p+1)!\lambda!(M-\lambda)!(N-\lambda+p)!(\lambda-p)!}\right]^{1/2}.
\end{aligned}
\tag{15}
$$

The collapsed wavefunction $|\Psi_p^{M,N}\rangle$ of Eq. 4 is given by Eq. 14, with the coefficients $E_\lambda$ as given in Eq. 15.

## A.2   RVB construction

We now analyze the RVB wavefunction introduced in the main text. We will show that this RVB state is, in fact, identical to the collapsed state. The rules for constructing the RVB state are shown in Fig. 2 of the main text. Note that, by construction, the RVB state has in-row permutation symmetry.

We decompose the RVB state into row wavefunctions,

$$|\Psi_{RVB}\rangle = \sum_{\kappa=p}^{M} F_\kappa |S_{tot} = M/2, m_{tot} = M/2 - \kappa\rangle_t \otimes |S_{tot} = N/2, m_{tot} = \kappa - p - N/2\rangle_b. \quad (16)$$

Each row is constrained to have the maximal $S_{tot}$ value ($M/2$ for the top row and $N/2$ for the bottom row) by in-row permutation symmetry. The $m_{tot}$ values are chosen to add to $m_{tot} = \left(\frac{M-N}{2}\right) - p$, appropriate for the RVB state as seen from Fig. 2. The coefficients $F_\kappa$ can be obtained as follows. We first expand the RVB state in the $S_z$ basis and regroup terms. We have

$$|\Psi_{RVB}\rangle = \frac{1}{\delta_{M,N,p} 2^{(M-p)/2}} \sum_{\kappa=p}^{M} (-1)^{\kappa-p} \begin{pmatrix} M-p \\ \kappa-p \end{pmatrix}$$

$$\left[ \sum_{P_M} | \underbrace{\uparrow \ldots \uparrow}_{M-\kappa} \underbrace{\downarrow \ldots \downarrow}_{\kappa} \rangle_t \right] \otimes \left[ \sum_{P_N} | \underbrace{\uparrow \ldots \uparrow}_{\kappa-p} \underbrace{\downarrow \ldots \downarrow}_{N+p-\kappa} \rangle_b \right]. \quad (17)$$

Here, $\delta_{M,N,p}$ is a normalization constant. The factor of $2^{-(M-p)/2}$ comes from the $(M-p)$ dimer singlet wave functions. To expand in the $S_z$ basis, we have taken $(\kappa-p)$ dimers to contribute $(-)|\downarrow_t \uparrow_b\rangle$ while the remaining $(M-\kappa)$ dimers contribute $|\uparrow_t\downarrow_b\rangle$. This leads to $\kappa \downarrow$ spins in the top row, with the index $\kappa$ running from $p$ to $M$. The choice of $(\kappa-p)$ dimers can be made in multiple ways – this gives rise to the factor of $\begin{pmatrix} M-p \\ \kappa-p \end{pmatrix}$.

We reexpress this in terms of row-wise angular momentum eigenstates. As each row is constrained to be symmetric under any permutation, the angular momentum states can be easily constructed to give

$$|S_{tot=M/2}, m_{tot} = M/2 - \kappa\rangle_t =$$

$$\frac{1}{(M-\kappa)! \kappa!} \begin{pmatrix} M \\ \kappa \end{pmatrix}^{-1/2} \left[ \sum_{P_M} | \underbrace{\uparrow \ldots \uparrow}_{M-\kappa} \underbrace{\downarrow \ldots \downarrow}_{\kappa} \rangle_t \right], \quad (18)$$

$$|S_{tot=N/2}, m_{tot} = \kappa - p - N/2\rangle_b = \frac{1}{(\kappa-p)!(N-\kappa+p)!}$$

$$\times \begin{pmatrix} N \\ \kappa-p \end{pmatrix}^{-1/2} \left[ \sum_{P_N} | \underbrace{\uparrow \ldots \uparrow}_{\kappa-p} \underbrace{\downarrow \ldots \downarrow}_{N-\kappa+p} \rangle_b \right]. \quad (19)$$

Comparing Eqs. 16,17,18 and 19, we have

$$F_\kappa = \frac{(-1)^{\kappa-p}}{\delta_{M,N,p} 2^{(M-p)/2}} \begin{pmatrix} M-p \\ \kappa-p \end{pmatrix} (M-\kappa)! \kappa! \begin{pmatrix} M \\ \kappa \end{pmatrix}^{1/2} \times$$

$$(\kappa-p)!(N-\kappa+p)! \begin{pmatrix} N \\ \kappa-p \end{pmatrix}^{1/2}. \quad (20)$$

We now compare Eqs. 14 and 16 which provide row wise decompositions of the collapsed state and the RVB state respectively. By choosing

$$\delta_{M,N,p} = \sqrt{\frac{M!N!(M-p)!(N-M+p)!p!(N+p+1)!}{2^{M-p}(N-M+2p+1)!}},$$ (21)

we find that the coefficients $F_\lambda$ and $E_\lambda$ become identical! This signifies that the collapsed state is indeed the RVB wavefunction that we have constructed.

# B  Probability distribution of emitted photons

In the main text, we discuss the probability distribution for photon emission from the Dicke model with a lossy cavity. Using Clebsch Gordan coefficients, Eq. 9 gives an expression for the probability distribution,

$$
\begin{aligned}
P_{N,M}(p) &= \frac{M!N!(1+2p+N-M)}{(M-p)!(1+N+p)!}. \\
&= \left[ \prod_{\xi=0}^{p-1} \frac{M-\xi}{N+\xi+1} \right] \frac{1+2p+N-M}{N+p+1}.
\end{aligned}
$$ (22)

We have cancelled common terms in the factorials to arrive at this form. The index $\xi$ runs over integers from 0 to $p-1$. We exponentiate the product to give

$$
\begin{aligned}
P_{N,M}(p) &= \frac{1+2p+N-M}{N+p+1} \exp\left[ \sum_{\xi=0}^{p-1} (\ln\{M-\xi\} - \ln\{N+\xi+1\}) \right] \\
&= \frac{\frac{1}{M}+2\gamma+\alpha-1}{\alpha+\gamma+\frac{1}{M}} \times \exp\left[ M\frac{1}{M} \sum_{\zeta=0}^{\gamma-\frac{1}{M}} (\ln\{1-\zeta\} - \ln\left\{\alpha+\zeta+\frac{1}{M}\right\}) \right].
\end{aligned}
$$ (23)

We have divided out by $M$ in all terms. We have introduced a new index $\zeta$ which increases in steps $1/M$. We have also multiplied and divided by $M$ in the exponent to facilitate the next step where we will take $M$ to be large and convert the summation into an integral. We have

$$
\begin{aligned}
P_{N,M}(p) &\approx \frac{\frac{1}{M}+2\gamma+\alpha-1}{\alpha+\gamma+\frac{1}{M}} \\
&\times \exp\left[ M \int_0^\gamma d\zeta \left( \ln\{1-\zeta\} - \ln\{\alpha+\zeta+\frac{1}{M}\} \right) \right].
\end{aligned}
$$

The integral in the exponent can be performed to obtain an analytic expression. However, the result cannot be easily interpreted. Instead, we approximate this expression assuming that $M$ is large and $\alpha$ is close to unity, i.e., our initial state has a small imbalance if at all.

As $\alpha \sim 1$, we can ignore the $1/M$ in the denominator of the prefactor. We can neglect $1/M$ in the numerator, assuming $\gamma \gg 1/M$, i.e., $p \gg 1$: we are interested in emission of a significant number of photons. At the same time, when $\alpha \sim 1$, we find numerically that $P(\gamma)$ is non-negligible only when $\gamma$ is small. On this basis, we take the integration variable $\zeta$ to be small, allowing us to Taylor expand the logarithms in the integrand.

$$P(\gamma) \approx \frac{2\gamma + \alpha - 1}{\alpha + \gamma} \exp\left[M \int_0^\gamma d\zeta \left(1 - 2\zeta - \alpha - \frac{1}{M}\right)\right]$$
$$= \frac{2\gamma + \alpha - 1}{\alpha + \gamma} \exp\left[-M\gamma^2 + M(1-\alpha)\gamma - \gamma\right].$$

We now consider three cases and appropriately simplify these expressions.

## B.1  Towards lesser emission: $\alpha > 1$

In this case, we have more up-spins than down-spins in the initial state. We focus on emission that is smaller than the imbalance in the initial state, i.e., we focus on $\gamma < (\alpha - 1)$. Assume that $\gamma$ is small, we have

$$P_{\alpha>1}(\gamma) \approx (\alpha - 1)\exp\left[-M(\alpha-1)\gamma\right]. \tag{24}$$

We see that the distribution is peaked at $\gamma = 0$ and exponentially decays with width $\{M(\alpha-1)\}^{-1}$. The numerical form of the probability distribution is consistent with this form when $M$ is taken to be large.

## B.2  Balanced case: $\alpha = 1$

Taking $\alpha$ to be precisely unity, we rewrite the distribution assuming $\gamma$ to be small. With $\delta = 0$, the probability distribution in Eq. 24 simplifies to

$$P_{\alpha=1}(\gamma) \approx \frac{2\gamma}{1+\gamma}\exp\left[-M\gamma^2\right]. \tag{25}$$

This is reminiscent of the expression for black body radiation. At large $\gamma$ ($\gamma \gg 1/\sqrt{M}$), this distribution decays exponentially. At small $\gamma$ values, it increases linearly. This suggests that this function has a maximum that can be obtained by extremizing with respect to $\gamma$,

$$\left.\frac{\partial P_{\alpha=1}(\gamma)}{\partial \gamma}\right|_{\gamma_m} = 0 \implies \gamma_m \sim \sqrt{\frac{1}{2M}}.$$

From the form of the exponential, we see that it has standard deviation $\sim M^{-1/2}$. This is borne out by a numerical examination of the distribution.

## B.3  Towards higher emission: $\alpha < 1$

In this case, we have more down-spins in the initial state. A crucial difference emerges from the $\alpha > 1$ case discussed earlier. Here, there is a lower bound for photon emission with atleast $M - N$ photons being emitted. This translates to a lower bound on $\gamma$, $\gamma_c = 1 - \alpha$. For $\gamma < \gamma_c$, the probability distribution $P(\gamma)$ is uniformly zero. To get the form of the distribution close to the threshold value, we redefine $\gamma = \gamma_c + \delta\gamma$, assuming $\delta\gamma \ll 1$. We obtain

$$P_{\alpha<1}(\gamma) \approx \frac{(\gamma_c + 2\delta\gamma)}{1+\delta\gamma}\exp\left[-M\gamma^2 - M\gamma_c\gamma\right]$$
$$\approx \gamma_c \exp\left[-M(\gamma_c + \delta\gamma)(2\gamma_c + \delta\gamma)\right]$$
$$\approx \gamma_c \exp\left[-2M\gamma_c^2\right]\exp\left[-3M\gamma_c\delta\gamma\right]. \tag{26}$$

We see that the distribution decays with increasing $\delta\gamma$. Clearly, it is peaked at $\gamma = \gamma_c$ and decays exponentially with width $\sim M^{-1}$. This is consistent with the numerically obtained distribution when $M$ is large.

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
