# Peer review of "Resonating valence bonds and spinon pairing in the Dicke model"

_SciPost Physics, doi:SciPost Phys. 4, 044 (2018)_

## Round 1 · Referee Report · Indrani Bose · 2018-6-3

Strengths

1. Proposes a novel scheme to synthesize resonating valence bond (RVB) states using the Dicke Hamiltonian.
2. Simulation protocol suggested of spinon-doping in cavity QED experiment.

Weaknesses

1. The paper is interesting with no specific weakness.

Report

The RVB states are entangled many body wavefunctions which are of considerable relevance in understanding the physical properties of strongly correlated systems. In these states spin pairs form singlets. The proposal put forward by the Authors to synthesize RVB states in a cavity QED setup utilising the Dicke model of two-level atoms, represented as S =1/2 spins, coupled to a common photon mode is novel and of significant interest. A spinon-doped RVB state is obtained when a Stern-Gerlach measurement of emitted photons is made with the spinons representing unpaired spins. An analogue of a finite-sized superconductor is obtained in the form of a mixed state with fluctuating spinon pairs under non-measurement of photons. The protocol when extended to large system sizes demonstrates a quantum phase transition in the open Dicke model. The proposal of using the Dicke system to synthesize RVB states has advantages over solid state and cold atom experiments.

Requested changes

None.

---

## Round 1 · Referee Report · Anonymous · 2018-6-5

Strengths

1. It present a novel scheme of detecting RVB physics using an optical setup.

2. It present a rigorous mathematical analysis of the phenomenon in the context of the experimental setup.

Weaknesses

1. The paper makes a set of assumptions which would make experimental realizations of the theory tricky.

Report

The paper proposes a novel scheme for realizing RVB state using optical (lossy cavity) setup. The paper is well written and easy to understand and I would recommend publication of the following changes are made ( see below)

Requested changes

1. One needs a clear discussion supplemented by explicit numbers about the assumptions. For example, the assumptions are listed in the paper. Form each of these, it would be nice to have numerical estaimate of several experimental parameters which could refklect the possibility of exterimental validity of the assumtpion.

2. Coudl the author elaborate a bit more, at a qualitative level, as to what happens to the realization of spinon physics in case each of the assumtpions are relaxed. When would one expect the effect that they want to see go away?

---

## Round 2 · Author Response

We thank the referees for their comments and the editor for the recommendation. In line with the comments sought by Referee 2, we have included some parameters from a suitable experimental setup. We believe that the modified manuscript makes a more convincing case for the viability of our proposals.

We are happy to see that the first referee has recommended publication as is. The second referee finds our work to be novel and well written. He/she has raised two questions which are important from the point of view of achieving an experimental realization. We ourselves have been thinking about these questions for some time.

The first question is about numerical estimates to support the assumptions made in our work. In the revised manuscript, we have included experimental parameters from Mlynek et al. (2014), an experimental study of two superconducting qubits in a microwave cavity. The most important assumption in our work is the ‘lossy cavity’ limit. These parameters show that the lossy-cavity limit is indeed experimentally realisable. In addition, the numbers also justify the use of the rotating wave approximation. We have modified the final section (Summary and Discussion) to discuss these parameters and their relevance.

For other assumptions in our work, it is not easy to obtain numerical estimates. For example, we are not aware of any experiments which have estimated the spin-spin interaction energy scale within a lossy-cavity setup. Such quantities are highly sensitive to details of the experimental setup and cannot be generically estimated. If an experimental lab were to take up our proposals, we believe they can make straightforward consistency checks to see if our assumptions are satisfied (e.g., the fidelity with which a singlet dark state is obtained in a two spin system).

In the second question, the referee asks about the consequences that arise when our assumptions are violated. As the referee has noted, the manuscript clearly lays out its assumptions. Weak violations of assumptions will lead to weak deviations in results, e.g., the probabilities for emission of different photon numbers could be altered by small amounts. In the manuscript, we have speculated that spin-spin interactions (neglected in our approach) could bring about a qualitative change to induce coherent Cooper pairing. We also have speculated that these effects can lead to a preference for shorter dimers. A more precise discussion is beyond the scope of the manuscript as there are a large number of possibilities. We hope to soon present a quantitative analysis, broadly discussing the experimental viability of our proposal. For instance, we hope to provide estimates for emission-time, a quantity that is central to one of our assumptions. For the sake of brevity, we believe it is best to keep this manuscript at the level of a theoretical proposal that is rigorous as long as the stated assumptions are obeyed.

We hope to convince the referee (and readers) that our manuscript proposes new directions, and will motivate further theoretical and experimental work.

---

## Round 2 · List of Changes

We have made several minor changes for grammar and for clarity.

We have included a new paragraph ('With a suitable experimental system such as...') in the 'Summary and Discussion' section. This provides parameters from an experimental paper -- we argue that the realized setup is favourable for our proposals.

---

## Editorial Decision

published